Interannual and spatial variability of maple syrup yield as related to climatic factors

Duchesne Louis 1 louis.duchesne@mrn.gouv.qc.ca
Houle Daniel 1 2
1 Direction de la recherche forestière, Ministère des Forêts, de la Faune et des Parcs , Québec , Canada
2 Consortium sur la climatologie régionale et l’adaptation aux changements climatiques (Ouranos) , Montréal, Québec , Canada
Miao Chiyuan
Electronic publication date: 2014 Jun 10
Publication date: 2014
Volume: 2
Electronic Location ID: e428
Received 2014 Apr 9; Accepted 2014 May 26
Copyright: © 2014 Duchesne and Houle
Copyright year: 2014
Copyright holder: Duchesne and Houle
License: This is an open access article distributed under the terms of the Creative Commons Attribution License, which permits unrestricted use, distribution, reproduction and adaptation in any medium and for any purpose provided that it is properly attributed. For attribution, the original author(s), title, publication source (PeerJ) and either DOI or URL of the article must be cited.
License URL: https://creativecommons.org/licenses/by/4.0/

Keywords: Maple syrup, Sugar maple, Climate, Cold hardiness, Acer saccharum, Freeze–thaw events, Yield statistics

Funding: Ministère des Ressources naturelles du Québec 112310065 This research was supported by the Ministère des Ressources naturelles du Québec, project number 112310065. The funders had no role in study design, data collection and analysis, decision to publish, or preparation of the manuscript.

==============================
Sugar maple syrup production is an important economic activity for eastern Canada and the northeastern United States. Since annual variations in syrup yield have been related to climate, there are concerns about the impacts of climatic change on the industry in the upcoming decades. Although the temporal variability of syrup yield has been studied for specific sites on different time scales or for large regions, a model capable of accounting for both temporal and regional differences in yield is still lacking. In the present study, we studied the factors responsible for interregional and interannual variability in maple syrup yield over the 2001–2012 period, by combining the data from 8 Quebec regions (Canada) and 10 U.S. states. The resulting model explained 44.5% of the variability in yield. It includes the effect of climatic conditions that precede the sapflow season (variables from the previous growing season and winter), the effect of climatic conditions during the current sapflow season, and terms accounting for intercountry and temporal variability. Optimal conditions for maple syrup production appear to be spatially restricted by less favourable climate conditions occurring during the growing season in the north, and in the south, by the warmer winter and earlier spring conditions. This suggests that climate change may favor maple syrup production northwards, while southern regions are more likely to be negatively affected by adverse spring conditions.

Introduction

Sugar maple (Acer saccharum Marsh.) is broadly distributed in North America, throughout the northeastern United States and southeastern regions of Canada (Little, 1971). Due to the exceptional sweetness of its sap, this species is commercially exploited for maple syrup production. In 2011, the world production of maple syrup was estimated at 49.5 millions litres, 79% of which were produced in Canada, and 21% in the U.S. (FPAQ, 2012).

The relation between the timing of the period of sapflow, syrup yield, and the fluctuation of air temperatures around 0 °C is intuitively known by producers; it has been observed and characterised in scientific studies, including Marvin & Erickson (1956), Plamondon (1977), and Pothier (1995). However, the climatic conditions prevailing during the sapflow season are not the only determinants of yield. Previous winter conditions, such as high mean January temperatures, can also be negatively correlated to yield, possibly due to the importance of cold hardiness on sap sugar content (Rock & Spencer, 2001; Duchesne et al., 2009; Tyminski, 2011). Along with local climatic conditions, many other factors can affect maple syrup yield, including physical tree parameters (Blum, 1973), genetic characteristics (Kriebel, 1989), foliar chemistry (Leaf & Watterston, 1964), soil fertility (Watterston, Leaf & Engelken, 1963) and sap extraction and conversion methods (Morrow & Gibbs, 1969). Although these factors can explain variations in maple syrup yield between trees or sites, climate remains the main factor affecting annual yield fluctuations, through its effect on sap flow fluxes, sugar concentration, or both (Marvin & Erickson, 1956; Cool, 1957; Pothier, 1995; Duchesne et al., 2009).

The importance of the maple syrup industry has increased considerably over the last decades. Technological developments have improved efficiency and allowed individual farms to exploit significantly more taps in their maple stands. Nevertheless, climate variability induces high year-to-year fluctuations in production volume, which often leave maple syrup producers with low-yield years, during which the benefit-cost ratio is reduced.

Some recent studies suggest that the maple syrup industry may be significantly affected by climate change. Duchesne et al. (2009) coupled the results of a regression model that linked the annual yield of Quebec’s industry to climate over a 22-year period with a dataset of future climatic scenarios obtained from several global climate models driven by different scenarios of CO2 emissions. Their results indicate that maple syrup yield could experience a 15% decrease by 2050 and a 22% decrease by 2090, compared to the 1984–2006 reference period. Nevertheless, assuming that the variables included in the prediction model reveal a pattern of climatic conditions that could occur earlier in the season, total maple syrup yield could be maintained at its current level if the period of sap production was advanced by 12 days in 2050, and by 19 days in 2090. From a simple model of sapflow potential based on biology and physics, Skinner, DeGaetano & Chabot (2010) reach similar conclusions for the U.S. industry; they predict that the warmer winter temperatures expected for the twenty-first century will result in a decline in the number of sapflow days, if the traditional sap collection schedules are maintained. They also suggest that the number of sapflow days across the northeastern U.S. might be maintained if the current sap collection schedule was adapted to occur 19 to 30 days earlier (depending on the climate scenario used) by 2100. Other impact assessment reports also highlight the potential effects of climate warming on the maple syrup industry (Rock & Spencer, 2001; Maclver et al., 2006; Frumhoff et al., 2007).

Studies documenting the effect of climate on maple syrup yield are based on previously established relationships between sapflow and the daily alternation of freezing and thawing temperatures for specific sites (Maclver et al., 2006; Frumhoff et al., 2007; Skinner, DeGaetano & Chabot, 2010), or on regression models linking variability in climate to the interannual average maple syrup yield at different regional scales (Rock & Spencer, 2001; Duchesne et al., 2009; Tyminski, 2011). Nevertheless, none of these studies document the simultaneous influence of climate on maple syrup yield on both interannual and interregional variability. Such information is a prerequisite for evaluation of the impact of climate change on the maple syrup industry over a broad territory. For instance, a site effect could be due to average climatic conditions (temperature and precipitation) that influence general growth conditions, and indirectly, syrup yield. This effect cannot be ascertained by only looking at year-to-year yield variations at a given site.

The main objective of the present study was to identify the factors responsible for interregional and interannual variability in maple syrup yield in northeastern North America over the 2001–2012 period. We combined the data from 8 Quebec regions and 10 U.S. states covering a broad region in which mean annual temperature varies by 8.8 °C. We hypothesized (1) that interregional variations would be predicted by average regional climatic conditions, and (2) that both interregional and interannual variations would be predicted by climatic variables.

Materials and Methods

Maple syrup yield

Annual maple syrup yields over the 2001–2012 period were compiled from the National Agricultural Statistics Service (USDA-NASS, 2012) and the 2012 economic report of the Federation of Quebec Maple Syrup Producers (FPAQ, 2012). Average annual yield data for the 12-year period were available for 8 administrative regions in Québec and for 10 U.S. states (Fig. 1), for a total of 216 individual years of production. Yield statistics were reported in U.S. gallons per tap for the U.S. industry and in pounds per tap for Quebec. Data were converted to metric values (ml tap−1) using conversion factors of 11.03 pounds per U.S. gallon and 344 ml per pound (Agriculture and Agri-Food Canada, 2007).

Figure 1 Sugar maple distribution in northeastern North America (Little, 1971). The numbers correspond to the grouped administrative regions of Quebec (Canada) and American states with significant maple syrup production. 1, Capitale-Nationale/Saguenay–Lac-St-Jean; 2, Bas-St-Laurent–Gaspésie; 3, Centre-du-Québec/Mauricie; 4, Laurentides/Outaouais/Abitibi-Témiscamingue; 5, Lanaudière/Laval/Montréal; 6, Chaudière-Appalaches; 7, Estrie; 8, Montérégie; 9, Maine; 10, Vermont; 11, New Hampshire; 12, Wisconsin; 13, Michigan; 14, New York; 15, Massachusetts; 16, Pennsylvania; 17, Connecticut; 18, Ohio. White dots represent the 25 km × 25 km grid used to generate climate data.

Meteorological data

Daily precipitation and temperature were generated for each state or region using the BioSIM model (Régnière, 1996; Régnière & St-Amant, 2007). BioSIM was originally developed to simulate insect development at the regional scale as a function of time-series of weather data. It operates by matching the geo-referenced sources of weather data (120 monitoring stations including the last 30 years of data) to the specified location, and then adjusts the selected sources of weather data to the specified latitude, longitude, elevation, slope, and aspect. The correlation between estimated and measured values is generally over 98% (Régnière & Bolstad, 1994).

Daily temperature and precipitation data were generated for each intersection point on a 25 km × 25 km grid covering the study area within the overall distribution of sugar maple (Fig. 1). Daily climate data were then averaged by state and region.

Influence of climate on maple syrup yield

We investigated the linear relationships between a set of selected climate variables and interregional and interannual maple syrup yield variability using a multiple regression model (1). (1) Yi=β0+β1xi1+β2xi2+⋯+βpxip+εi

where xij is the ith observation on the jth independent variable, β0 the model intercept, β’s the model parameters, and εi the error term. Climate variables were chosen based on results of previous studies, which identified the most influential climate variables for maple syrup production (see introduction). We focussed our analysis mainly on variables characterizing daily freeze–thaw events, winter hardiness, and climatic conditions that prevailed during the previous growing season. The variables tested are listed in Table 1. In addition to these, time was also tested to investigate potential annual trends in yield that might not be explained by climate (e.g., yield improvements associated to technological developments, or yield decline due to overtapping). Similarly, a potential difference between countries (Canada vs the U.S.) was also assessed to take into account the possible effect of methodological differences in the acquisition of yield statistics (e.g., weight vs volume measurements, survey protocols, etc.). All possible regression models were tested with the RSQUARE procedure (SAS Institute, 2002) to determine the maximum amount of variance in maple syrup yield that could be explained as a function of climate variables. The final models were selected based on Mallow’s Cp-statistic (Mallows, 1973) and Akaike’s information criteria (Akaike, 1973). Multicollinearity among climate variables of the selected models was tested using condition indexes and the variance inflection factor, to verify that dependencies among variables did not affect the regression estimates (Belsey, Kuh & Welsch, 1980). Residuals were graphically checked to assess the tenability of the regression assumptions. Partial regression residual plots, factoring out the influence of other covariables, were used to summarize the relationships between climate and maple syrup yield.

Table 1 Selected climate variables used to model yearly regional maple syrup yield.

Variables	Mean	Min.	Max.	
Previous summer conditions				
Total heating degree days >3 °C (°C)	2,398	1,506	3,611	
Maximum temperature (°C)	32	26	36	
Total precipitation during days
with mean temperature >3 °C (mm)	807	459	1,493	
Previous winter conditions				
Total cooling degree days <−1 °C (°C)	737	48	1,776	
Minimum temperature (°C)	−26	−41	−14	
Total precipitation during days
with mean temperature <−1 °C (mm)	203	18	476	
Freeze–thaw events and spring conditions				
Frequency of daily freeze–thaw events (days)	32	12	56	
Minimum temperature
during the freeze–thaw events (°C)	−13	−7	−24	
Maximum temperature
during the freeze–thaw events (°C)	16	10	24	
Mean temperature
during the freeze–thaw events (°C)	1.5	0.9	3.3	
Mean daily temperature range
during the freeze–thaw events (°C)	12	10	15	
First day of year when total heating
degree days reach 75 (DOY)	110	68	143	
Total precipitation during
the freeze–thaw events (mm)	62	7	225	
Notes.

A daily freeze–thaw event was recorded when the minimum temperature fell below −1 °C and the maximum temperature exceeded 3 °C.

Results

Variability in maple syrup yield from 2001 to 2012

Figure 2A shows the distribution of the individual observations of annual maple syrup yield, which ranged from 314 ml tap−1 (Wisconsin in 2012) to 1,306 ml tap−1 (Vermont in 2011), for an overall average (±1 s.e.) of 740 ± 11 ml tap−1. Mean annual yields ranged from 624 ± 20 ml tap−1 in 2001 to 951 ± 31 ml tap−1 in 2011 (Fig. 2B), while the regional yields varied from 640 ± 45 ml tap−1 in Pennsylvania (region 16) to 910 ± 28 ml tap−1 in Vermont (region 10; Fig. 2C).

Figure 2 Quantile plot of maple syrup yield for the 216 observations (A), and mean maple syrup yield averaged by year (B) and by region (C) over the time period on the studied territory. Error bars represent ±1 standard error (see Fig. 1 for regional boundaries and their numbering).

Air temperature and precipitation variability from 2001 to 2012

For individual years and regions, mean annual temperature (Fig. 3A) varied from 0.9 °C (Capitale-Nationale/Saguenay–Lac-St-Jean in 2004) to 12.3 °C (Ohio in 2012), for an overall average of 6.4 ± 0.2 °C. Yearly averaged mean annual temperature over the entire study area ranged from 5.4 ± 0.6 °C in 2003 to 7.5 ± 0.7 °C in 2012 (Fig. 3B), while regional averages over the whole period ranged from 2.2 ± 0.3 °C for Capitale-Nationale/Saguenay–Lac-St-Jean to 11.0 ± 0.2 °C for Ohio (Fig. 3C).

Figure 3 Quantile plot of mean annual temperature and annual precipitation for the 216 observations (A), and annual temperature (dots) and precipitation (bars) averaged by year (B) and by region (C), over the time period on the studied territory. Error bars represent ±1 standard error (see Fig. 1 for regional boundaries and their numbering).

Total precipitation ranged from 716 mm (Wisconsin in 2003) to 1,751 mm (Connecticut in 2011), with an overall average of 1,138 ± 14 mm (Fig. 3A). Yearly averaged total precipitation over the entire study area ranged from 928 ± 15 mm in 2001 to 1,327 ± 59 mm in 2011 (Fig. 3B). Regional averages over the whole period ranged from 828 ± 26 mm in the western edge of the range in Wisconsin (region 12) to 1,332 ± 59 mm for Connecticut (region 17; Fig. 3C).

Relationship between climate and maple syrup yield

The best multiple linear regression model using regional averages of meteorological variables, a time and a country effect, explained 44.5% of the variance in maple syrup yield records (Fig. 4) (2). (2) Yieldi∧=−36792+17.9×Yearsi+65.3×Countryi+0.26×GDDi−0.18×PPTi−13.0×MinTi+5.3×FFTi−16.9×MaxTFTi+48.5×MeanTFTi+6.6×DOY75i.

The climate variables include cumulative growing degree days (GDD, Fig. 5C) and precipitation during the previous growing season (PPT, Fig. 5D), daily minimum temperature (MinT, Fig. 5E), frequency of spring freeze–thaw events (FFT, Fig. 5F), maximum (MaxTFT) and mean temperatures (MeanTFT) during the freeze–thaw events (Figs. 5G and 5H), and the day of year when the cumulative growing degree days reach 75 (DOY75, Fig. 5I). A portion of the variance that could not be explained by the climate variables was associated with temporal (Years, Fig. 5A) and regional effects (Country, Fig. 5B). The country effect is calculated based on a binary variable having the value of 1 for Canada and 2 for the U.S.

Figure 4 Observed and predicted yearly maple syrup yield at the regional scale.

The model explains (R2) 44.5% of the variance in maple syrup yield, with a mean absolute error of 61.8 ml tap−1 and a slight negative bias of 0.05 ml tap−1.

Figure 5 Partial regression residual plots summarising the effect of the selected climate variable on yearly regional maple syrup yield.

Together, these variables explain 44.5% of the variance in maple syrup yield with a mean absolute error of 61.8 ml tap−1. The variance inflation factors (< or = 5.2) and condition indexes (< or = 5.8) demonstrate the lack of problematic collinearity among independent variables. Parameter estimates along with their t-statistics and the associated probability (p-value) are provided for each variable included in the model.

Discussion

Maple syrup yield and average regional climate variability

Contrary to our first hypothesis, regional maple syrup yield variability was not related to the mean regional climatic conditions within the study area, despite the strong south-north gradient in average annual temperature (difference of 8.8 °C between region 1, Ohio, and region 18, Capitale-Nationale/Saguenay–Lac-St-Jean). Similarly, the particularly dry precipitation regimes (including winter precipitation) observed in Wisconsin (828 mm/yr, region 12) and Michigan (850 mm/yr, region 13) did not translate into a marked difference in maple syrup yield for these regions. These observations suggest that variability in annual maple syrup yield at the regional scale results from complex interactions with climate, and that the climate gradient alone cannot explain variability over the studied territory. For instance, higher annual average temperature in the southern portion of the study area may favor growth and vigor of sugar maple trees, which could positively affect syrup yield. However, higher mean temperatures could also result in less favorable springtime conditions, if temperatures are too high or if there are fewer freeze–thaw cycles. Average climatic conditions may reflect different realities along the gradients of mean annual temperature and precipitation observed over the studied territory. These considerations are discussed below. A potential effect of the climatic gradient on yield might also be obscured by regional variations in growth and vigor of maple trees, soil fertility, atmospheric acid deposition, and sap extraction and conversion methods.

Model of maple syrup yield

A multiple regression model using regional averages of climate variables and the effects of time and country explained 44.5% of the interannual and interregional variability in maple syrup yield for 18 regions over the 12-year period (n = 216). The Quebec and U.S. data are well distributed within the range of predicted and observed values (Fig. 4). The variance explained by our model is comparable to that of Tyminski’s (2011) model, which accounts for 44% of the interannual maple syrup yield variability observed in the state of New York over the 1916–2006 period. However, this is considerably less than the variance explained (84%) by the model of Duchesne et al. (2009) for the total yield in the province of Quebec over the 1985–2006 period, for which the interregional variability was somehow buffered at the provincial level. The model in the present study combines climate variables measured prior to the sapflow season (previous growing season and winter) and during the sapflow season itself, as well as terms accounting for yield variations over time and between countries that cannot be explained by the climate variables.

Impacts of the previous growing season and winter conditions on yield

Our model suggests that the maple syrup yield of a given sapflow season is significantly affected by the climatic conditions prevailing during the previous growing season. Warmer and drier conditions during a given growing season were associated to a higher yield during the subsequent period of sapflow (Figs. 5C and 5D). Maple syrup yield was much more sensitive to temperature than to precipitation: a variation of 2,105 cumulative growing degree days (from 1,506 to 3,611) resulted in a yield variation of 545 ml tap−1, while a variation of 1,034 mm in precipitation (from 459 to 1,493 mm) was associated to a yield variation of only 183 ml tap−1 during the subsequent sapflow period (Fig. 5D). Warmer summer conditions are associated with higher maple syrup yield (Fig. 5C). Dendroclimatic studies have shown that warm summer temperatures generally favour greater radial growth rates of sugar maple trees (Payette, Fortin & Morneau, 1996; Goldblum & Rigg, 2005). Although there is no clear demonstration of a causal link between sugar maple growth rate and maple syrup yield, a number of studies have suggested positive relationships between sugar maple growth and sap sugar content, sapflow amounts, or both. For example, Laing & Howard (1990) showed a link between sap sugar content and growth rate, and suggested an association between low competition pressure and greater sap sweetness. In good agreement with this, Morrow (1955) reported that maple trees with large crowns tended to produce more and sweeter sap than suppressed trees or those that were subjected to competition. Morselli, Marvin & Lang (1978) observed that maples with sweeter sap had larger xylem rays than did trees whose sap was less sweet. All of these observations concur with our results and support the hypothesis that the previous growing season conditions influence maple syrup yield, probably due to the positive influence of climate on tree growth and starch storage capacity (Tyminski, 2011). We presume that the weak and negative correlation between precipitation levels during the previous growing season and maple syrup yield reflects the negative impact of reduced photosynthetically active radiation on tree growth, as a result of increased cloud cover (Goldblum & Rigg, 2005).

Climate during the previous winter also affects yield. Winter cold hardiness accounts for a large part of the spatial variability in maple syrup yield; the colder territories in the north are favoured compared to those in the south (Fig. 5E). A progression in daily minimum temperature from −40.7 to −13.6 °C resulted in a yield variation of 353 ml tap−1. In the New England Regional Assessment Group report, Rock & Spencer (2001) provide data showing a moderate negative correlation between total U.S. syrup production and mean winter temperature between 1916 and 1999. Duchesne et al. (2009) and Tyminski (2011) also observed that yearly variations in maple syrup yield in the province of Québec and the state of New York were negatively correlated to mean January temperatures. In various types of plant cells, sugar content appears correlated to frost hardiness (Sakai, 1960). Sugar acts as a biological antifreeze in plants (Sauter & Van Cleve, 1991; Yuanyuan et al., 2009). The accumulation of soluble sugar in roots of sugar maple trees is known to coincide with periods of lowest soil temperatures; this suggests that winter temperature plays a decisive role in the sugar–starch–sugar cycle (Bertrand et al., 1999). Freezing tolerance of plants can also be enhanced by a gradual exposure to low temperatures. Soluble sugars play an important role in this process, known as cold acclimation (Yuanyuan et al., 2009). Consequently, we hypothesize that the negative correlation observed in our dataset between maple syrup yield and daily minimum winter temperatures is the result of sugar maple cold acclimation.

Freeze–thaw events and spring conditions

Skinner, DeGaetano & Chabot (2010) suggested that there was a well-established and predictable relationship between maple sapflow and alternating freezing and thawing temperatures. This conclusion is based on unpublished data from Eggleston and Chabot showing that sapflow recorded over a 30-year period at the Uihlein Forest (Lake Placid, NY) occurred on 80% of the days when the minimum temperature fell below −1.1 °C and maximum temperature exceeded 2.2 °C. Similarly, Plamondon (1977) reported that the daily minimum temperature and the difference between the minimum and maximum temperatures were correlated with daily sapflow in a one-year experiment, while Pothier (1995) reported that annual sap yield and sap sugar concentration over a 15-year record were strongly correlated to the number of days in spring during which temperatures fluctuated around 0 °C. On the other hand, Cool (1957) found that daily maximum temperature was the only climatic factor significantly correlated with sap flow, Kim & Leech (1985) found that maximum temperature was the most important climatic factor explaining increases in daily sap flow over a 5-year period, and Marvin & Erickson (1956) showed that freezing temperatures were not a necessary prerequisite for sapflow. In the present study, both the frequency of freeze–thaw events and the mean temperature during these events were positively related to maple syrup yield, while the maximum daily temperature over the same period was negatively related to maple syrup yield (Figs. 5F–5H). The frequency of freeze–thaw events and the associated temperatures appear to capture mainly interannual variability in maple syrup yield. However, these variables explain much less spatial variability than the climatic conditions prevailing during the previous growing season and winter. Consequently, although previous studies have shown that sapflow was correlated with the frequency of freeze–thaw events at a given site or for a given territory, our results suggest that this variable alone does not capture the interregional variability in maple syrup yield.

The day of year when the cumulative growing degree days reached 75 was strongly and positively related to maple syrup yield, accounting for a large part of its spatial variability. A variation of 75 days resulted in a yield variation of 493 ml tap−1 (Fig. 5I). In the southern portion of the study area, early spring conditions had a more negative impact on maple syrup yield than did late spring conditions occuring in the north. High air temperatures not only induce bud break and interrupt sweet sap flow, but also accelerate the physical plugging of the tap, which is caused by the combined effects of a bacterial invasion and vessel blockage by microorganisms, gummy substances, or both (Ching & Mericle, 1960).

Trend in time and intercountry differences

Within the total explained variance (44.5%), a portion was due to an increasing temporal trend in maple syrup yield (7.7%) and to intercountry differences (0.4%) which were not related to a concomitant trend in the selected climate variables. The increasing temporal trend in yield contrasts with observations by Duchesne et al. (2009) who noted no yield increase over time (1985–2006), despite the efforts invested by producers over recent decades to improve existing industrial infrastructures. Efficient collection systems, including plastic tubing (Koelling, Blum & Gibbs, 1968) and vacuum pumping (Blum & Koelling, 1968) along with improved sanitation practices are known to increase total yield. These systems have gained in popularity over recent decades, with an expected positive effect on maple syrup yield. We also hypothesize that the increasing trend of maple syrup yield over time can be related to the addition of new, more productive taps. The number of exploited taps in Québec and the U.S. increased from 40.4 million in 2001 to 52.6 million in 2012 (Fig. 6). Over the same period, the maple syrup yield increase (unexplained by the selected climate variables) was approximately 155 ml tap−1. This increase was significantly correlated to the number of exploited taps (r = 0.87, p < 0.001). When a tap is drilled into the tree, the wood tissues surrounding the tap become stained. Tapping into stained wood reduces sapflow (Smith, 1971). Some guides and regulations are now provided to prevent overtapping, but many older sugarbushes may have been overtapped. Consequently, producers often observe that newly tapped trees are generally more productive than trees that have been tapped for many years. Not only can wood compartmentalization reduce the yield of old taps, but more efficient, newly installed equipment may also contribute to increasing the yield of new taps compared to older ones. The stability of the maple industry resulting from the Quebec Federation of Maple Producers strategic reserve and the strengthening of the Canadian dollar in comparison to the U.S. dollar may have contributed to stimulating expansion (additional taps) and technological investment, thus increasing productivity.

Figure 6 Temporal evolution of the number of exploited taps in Quebec and the U.S. (black circles) and of the residual of maple syrup yield (white circles), summarising the effect of time on yearly yield, while factoring out the influence of climate.

We also observed that sugar maple yield on U.S. territory was, on average, slightly higher (65 ml tap−1) than in Quebec. Consequently, part of the interregional variability in maple syrup yield might be explained by other factors, such as soil fertility, atmospheric acid deposition, sap extraction and conversion methods, and sugarbush management. In addition, intercountry comparisons were also affected by differences in survey procedures. For example, the unit of measurement used by both countries (weight for Canada and volume for U.S.) requires a conversion that may cause differences in the evaluation of sugar maple production. Consequently, the differences reported between U.S. and Canada yields that are not related to climate should be interpreted with caution.

Conclusion

The interregional and interannual variability in maple syrup yield were studied over the entire range of the sugar maple syrup production zone in Quebec and the northeastern United States. A large part of the variance in syrup yield was explained by climate, and more specifically by the conditions prevailing during the previous growing season and the previous winter. The frequency of spring freeze–thaw events and temperature conditions during the sapflow period explained a smaller part of the variability in syrup yield, mainly related to interannual variability in yield, and were much less spatially discriminating. Within the climatic gradient present in the study area, warmer growing seasons in the south were associated to higher maple syrup yields (possibly due to a positive influence on tree growth), while in the north, higher yields were associated to colder winters (probably related to increased frost hardiness) and later spring conditions. Consequently, optimal conditions for maple syrup production appear to be spatially restricted in the north by the less favourable climate conditions during the growing season, and in the south, by the warmer winter and earlier spring conditions. These observations suggest that, at the regional scale, annual maple syrup yield variability results from complex interactions with climate. This variability cannot not be explained solely by the climate gradient over the territory studied, nor by the frequency of freeze–thaw events that characterize a given season. Consequently, climate change risk assessment for the maple syrup industries of northeastern North America remains a challenge.

Supplemental Information

Supplemental Information 1 Regional averages of daily climate and annual maple syrup yield

Click here for additional data file.

We would like to thank the anonymous reviewers and Timothy Perkins for their valuable comments and suggestions to improve the quality of the paper. We are also grateful to Denise Tousignant and Debra Christiansen Stowe for English editing.

Additional Information and Declarations

Competing Interests

Author Contributions

Daniel Houle is an employee of the Consortium sur la climatologie régionale et l’adaptation aux changements climatiques. We declare that we have no significant competing financial, professional or personal interests that might have influenced the performance or presentation of the work described in this manuscript.

Louis Duchesne conceived and designed the experiments, performed the experiments, analyzed the data, wrote the paper, prepared figures and/or tables, reviewed drafts of the paper.

Daniel Houle wrote the paper, reviewed drafts of the paper.

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
