# Peer review of "Interannual and spatial variability of maple syrup yield as related to climatic factors"

_PeerJ, doi:10.7717/peerj.428_

## Round 0.1 · original submission · Major Revisions

Two Reviewers have provided their comments about this manuscript. Reviewer 1 is very critical. If you wish to revise your manuscript, please take the referee comments fully into account and provide point-by-point responses with a full list of changes. Revisions in this category may involve substantial text changes, recalculations or new analyses in addition to more minor clarifications and corrections. In particular, you should put your analysis into context with previous studies by comparing your results with literature, and resolve the issue with gridded observations, which may diverge from real data in data sparse areas. Other comments of the reviewers are relevant as well and you should follow them.

Reviewer 1 ·

Basic reporting

The English writing and the grammar should be revised by a mother tongue (who I am not). Several sentences are unclear, e.g., lines 43-45, 51-54. And the authors use “climate data”, “weather data” and also “climatic data”, which I believe are referring to the same “data”.

Experimental design

This paper collects the maple syrup yield data in 8 Quebec regions in Canada and in 10 states in the U.S. from 2001 to 2012 and establishes a multi-regression model to reveal climatic and regional effects. Regarding the regional differences between Canada and the U.S., the authors did point out the protocol and measuring units besides temperature. However, they are too general and the readers cannot see their relationship with the maple syrup yield.

Validity of the findings

On one hand, the multi-regression model the authors keep mentioning has not been presented. The readers have no idea what does it look like. What’s the value of R2 and how can it account for 44.5% of the total variance in maple syrup yield. Only the partial regression plots between the yield and different factors, including year, region and climatic conditions are shown (Figure 5). However, it is unclear what do “β” and “t” represent on each plot.

On the other hand, the reasoning in “Discussions” (Section 4) lacks the support of data. For example, the authors concluded that “Warmer summer conditions had a more positive impact on maple syrup yield in the southern regions than in colder regions in the north (Figure 5C)” (Line 194-196). Based on the results shown in Figure 5C, the clusters for Canada and the U.S. had no significantly different gradients.

Additional comments

This paper collects the maple syrup yield data in 8 Quebec regions in Canada and in 10 states in the U.S. from 2001 to 2012 and establishes a multi-regression model to reveal climatic and regional effects. The objective is very ambitious, but the analyses and discussions seem untenable.

Reviewer 2 ·

Basic reporting

No Comments

Experimental design

No Comments

Validity of the findings

No Comments

Reviewer 3 ·

Basic reporting

Sugar maple mainly distributed in Canada and northeastern United States. The research presented in this manuscript involved extensive data mining and statistics to model the maple syrup yield across the Canada and northeastern United States. Besides the authors also discussed the impact of climate factors on maple syrup yield on both interannual and interregional variability.
This study would contribute to our understanding of the impact of climate factors on maple syrup yield and the impacts of climatic change on the maple syrup industry in the upcoming decades.
While there are several major issues should be addressed.

Experimental design

Title: "Interannual and spatial variability of maple syrup yield as related to climate". "climate" contains a lot of meaning, which side of climate should be emphasized ("climatic factors" for example) .

Materials and methods:
L.92 to L.99, why to use BioSIM software to compute meteorological data? Did the computed meteorological data been verified and how? Did the BioSIM software need different model parameters in different region (8 administrative regions in Québec and for 10 U.S. states in this study). Those information are very important for read to understanding this work, so, it should be expressed clearly in " Materials and methods " part.
L.101 to L.102: "We investigated the relationships between a set of selected climatic variables and interregional and interannual maple syrup yield variability". My question is does investigated " a set " is enough to represent of total study areas?
L.102 to L.104: "Weather variables were chosen based on results of previous studies, which identified the most influential climate variables for maple syrup production". Which " previous studies " have been referenced?

Validity of the findings

Results:
L.143 to L.144:" A multiple regression model using regional averages of meteorological variables, a time and a country effect, explained 44.5% of the variance in maple syrup yield ". The specific model of the "multiple regression model" should be expressed in the manuscript, and a validation of the model (Please see "http://en.wikipedia.org/wiki/Verification_and_validation" for the description of validation) should be done before it been used.

Table 1:
L.410 to L. 413: The number of each variables is needed.

Additional comments

Sugar maple mainly distributed in Canada and northeastern United States. The research presented in this manuscript involved extensive data mining and statistics to model the maple syrup yield across the Canada and northeastern United States. Besides the authors also discussed the impact of climate factors on maple syrup yield on both interannual and interregional variability.
This study would contribute to our understanding of the impact of climate factors on maple syrup yield and the impacts of climatic change on the maple syrup industry in the upcoming decades.
While there are several major issues should be addressed.

Title: "Interannual and spatial variability of maple syrup yield as related to climate". "climate" contains a lot of meaning, which side of climate should be emphasized ("climatic factors" for example) .

Materials and methods:
L.92 to L.99, why to use BioSIM software to compute meteorological data? Did the computed meteorological data been verified and how? Did the BioSIM software need different model parameters in different region (8 administrative regions in Québec and for 10 U.S. states in this study). Those information are very important for read to understanding this work, so, it should be expressed clearly in " Materials and methods " part.
L.101 to L.102: "We investigated the relationships between a set of selected climatic variables and interregional and interannual maple syrup yield variability". My question is does investigated " a set " is enough to represent of total study areas?
L.102 to L.104: "Weather variables were chosen based on results of previous studies, which identified the most influential climate variables for maple syrup production". Which " previous studies " have been referenced?

Results:
L.143 to L.144:" A multiple regression model using regional averages of meteorological variables, a time and a country effect, explained 44.5% of the variance in maple syrup yield ". The specific model of the "multiple regression model" should be expressed in the manuscript, and a validation of the model (Please see "http://en.wikipedia.org/wiki/Verification_and_validation" for the description of validation) should be done before it been used.

Table 1:
L.410 to L. 413: The number of each variables is needed.

·

Basic reporting

This paper is well suited to be part of the scholarly literature. The results and conclusions are well justified. There is a relative lack of information on the effects of climate on maple syrup yield. The article is well-written and clear, and provides a suitable level of introduction to the material. Existing literature is appropriately referenced.

Experimental design

The experimental design is appropriate. Sufficient information is presented to be both understandable and replicable.

Validity of the findings

The findings are statistically sound and robust. The conclusions, to the extent provided, are appropriate.

Additional comments

A few minor revisions might be made, but are not wholly necessary.

The authors might consider and mention the impacts of two additional factors that have affected syrup yield and production over the past 10-12 yrs. These are: A) improved sanitation practices which have resulted in very strong yield improvements and B) the stability of the maple industry resulting from the Quebec Federation of Maple Producers strategic reserve and the strengthening of the Canadian dollar in comparison to the U.S. dollar. Both of these have created a climate where maple production can produce healthy profits for maple producers, and the profits are large enough to warrant further expansion (additional taps) and investment in better technology to produce more syrup (higher yields per tap). I would be happy to chat with the authors about this if they wish.

---

## Round 0.2 · accepted · Accept

The authors had made great progresses according to the reviewers' comments. Hence, acceptation is what I recommend now.